# Dye Separation and Antibacterial Activities of Polyaniline Thin Film-Coated Poly(phenyl sulfone) Membranes

**DOI:** 10.3390/membranes11010025

**Published:** 2020-12-29

**Authors:** Javed Alam, Arun Kumar Shukla, Mohammad Azam Ansari, Fekri Abdulraqeb Ahmed Ali, Mansour Alhoshan

**Affiliations:** 1King Abdullah Institute for Nanotechnology, King Saud University, Riyadh 11451, Saudi Arabia; ashukla@ksu.edu.sa (A.K.S.); mhoshan@ksu.edu.sa (M.A.); 2Department of Epidemic Disease Research, Institute of Research and Medical Consultation, Imam Abdulrahman Bin Faisal University, Dammam 31441, Saudi Arabia; maansari@iau.edu.sa; 3Chemical Engineering Department, College of Engineering, King Saud University, Riyadh 11451, Saudi Arabia; falhulidy@ksu.edu.sa; 4K. A. CARE Energy Research and Innovation Center at Riyadh, Riyadh 11451, Saudi Arabia

**Keywords:** nanofiltration membrane, polyaniline thin film, separation enhancer, antimicrobial coating

## Abstract

We fabricated a nanofiltration membrane consisting of a polyaniline (PANI) film on a polyphenylsulfone (PPSU) substrate membrane. The PANI film acted as a potent separation enhancer and antimicrobial coating. The membrane was analyzed via scanning electron microscopy and atomic force microscopy to examine its morphology, topography, contact angle, and zeta potential. We aimed to investigate the impact of the PANI film on the surface properties of the membrane. Membrane performance was then evaluated in terms of water permeation and rejection of methylene blue (MB), an organic dye. Coating the PPSU membrane with a PANI film imparted significant advantages, including finely tuned nanometer-scale membrane pores and tailored surface properties, including increased hydrophilicity and zeta potential. The PANI film also significantly enhanced separation of the MB dye. The PANI-coated membrane rejected over 90% of MB with little compromise in membrane permeability. The PANI film also enhanced the antimicrobial activity of the membrane. The bacteriostasis (*B*_R_) values of PANI-coated PPSU membranes after six and sixteen hours of incubation with *Escherichia coli* were 63.5% and 95.2%, respectively. The *B*_R_ values of PANI-coated PPSU membranes after six and sixteen hours of incubation with *Staphylococcus aureus* were 70.6% and 88.0%, respectively.

## 1. Introduction

Membrane separation techniques for water and wastewater treatment have been increasingly utilized to address the global challenge of water scarcity. These techniques have intrinsic advantages, such as facile scalability, low environmental impact, and most importantly, continuous separation in combination with other conventional processes [1,2,3,4,5,6]. Membrane processes to treat water provide several benefits. However, the inherently hydrophobic nature of the polymers used for membrane preparation is a disadvantage [7,8,9,10]. Membranes are prone to fouling due to their hydrophobic nature, so more power is required to force water flow. Consequently, the membrane must be replaced frequently. Several strategies have been employed to mitigate fouling, including pretreatment of the feed water and hydrophilic modification of the membranes. Hydrophilic polymers and nanoparticles with functional groups have been used to modify membrane surfaces via various approaches, such as coating, grafting, and bulk modification. Surface coatings have recently generated significant interest as an attractive means of hydrophilic membrane modification [9,11,12,13,14,15,16,17,18,19]. Surface coatings offer several advantages, such as low cost and facile adoption. The stability of a coating can be improved by tailoring its thickness and enhancing crosslinking.

In this study, a polyphenylsulfone (PPSU) membrane was used as a substrate and coated with a polyaniline (PANI) membrane. The properties of PPSU membranes are superior to those of membranes prepared from other sulfones, such as polysulfone and polyethersulfone (PES) [20,21,22]. However, the intrinsically hydrophobic nature of PPSU is often viewed as a limitation. Hence, we deposited a thin PANI film on the surface of a PPSU membrane to enhance its separation performance. PANI is an intrinsically conductive polymer that has been applied as a hydrophilic modifier to develop membranes for various separation applications. These include water treatment, gas separation, and fuel cell applications [23,24,25,26,27,28,29]. PANI can be utilized as a bulk powder. It can also be used to cast films and fabricate nanoparticles or fibers. Its acid-base doping chemistry facilitates the construction of membranes with switchable surface wettability properties and tunable pore sizes [28,30,31]. PANI is an ideal membrane modifier, because it is both hydrophilic and insoluble in water. PANI systems have ion exchanging properties and are environmentally sustainable. In several recent studies, PANI and sulfonated PANI were used as additives to prepare hydrophilic PPSU membranes [32,33]. To the best of our knowledge, PANI has been investigated primarily as a bulk powder. However, it has not been utilized to cast films on PPSU substrate membranes to enhance their surface properties. We generated a PANI film on a PPSU substrate membrane via the oxidative polymerization of an aniline monomer. The PANI-coated PPSU membrane was then compared to an unmodified PPSU membrane in terms of hydrophilicity, surface charge, contact angle, antibacterial activity, and separation performance via the filtration of a methylene dye solution. 

## 2. Materials and Methods 

### 2.1. Materials 

Distilled aniline, hydrochloric acid (HCl, ACS reagent, 37%), *N*-methyl-2-pyrrolidone (NMP, >98%), and ammonium persulfate (NH_4_)_2_S_2_O_8_ (APS, reagent grade, 98%) were used as received from Aldrich, (Kenilworth, NJ, USA). Polyphenylsulfone (PPSU) (Ultrason P 3010) was supplied by BASF (Berlin, Germany). Polyvinylpyrrolidone (PVP, average MW 10,000), methylene blue (MW 373), and sodium azide (≥99.5%) were purchased from Aldrich (Kenilworth, NJ, USA). Deionized (DI) water was obtained from a Milli-Q water purification system.

### 2.2. Fabrication of PPSU Support Membrane

The PPSU support membrane was prepared using a phase inversion method as described previously [20]. Dried PPSU (17 wt.%) and 10 wt.% PVP were both dissolved in *N*-methyl-2-pyrrolidone with gentle stirring for 3 h at 60 ± 5 °C. When no air bubbles were visible in the homogeneous mixture, it was hand-cast onto a clean and dry glass plate to a thickness of 100 ± 3 μm using a blade. The glass plate was gently placed in a coagulation bath containing distilled water for phase inversion. Within one minute, the membrane floated to the surface of the water. The membrane was removed, rinsed with DI water, and stored in an aqueous solution of 0.2% sodium azide to prevent microbial contamination. 

### 2.3. Deposition of a PANI Thin Film on the PPSU Membrane

A PANI film was grown on the surface of the PPSU substrate via the chemical oxidation of monomeric aniline. APS served as the oxidant, and HCl was used as a doping agent. The PPSU membrane was immersed in DI water for 1 h to remove any residual chemicals, then taped to a clean glass plate. Monomeric aniline (0.2 mL) was added to 1 M HCl (5 mL), and the mixture was stirred at room temperature. APS (0.01 g) was dissolved in DI water (0.25 mL), and the solutions were combined and stirred for 20 s to ensure complete mixing. The mixture was then poured over the surface of the PPSU membrane. The film turned purple at the beginning of the polymerization process. The color of the film changed to green, which confirmed the formation of PANI as an emeraldine salt. The PANI film was rinsed with DI water to remove granular PANI, then dried and stored in a desiccator. The following steps were performed to deposit a uniform PANI film using a ‘pouring’ method. (1) A solution to be polymerized was poured over the surface of the PPSU membrane and kept for 25 s. (2) The temperature to polymerize aniline was 25 °C, and (3) the low concentration of monomeric aniline (0.2 mL) played a great role in depositing a uniform PANI film over the surface of the PPSU membrane. Most important, it was merely the first cycle of PANI film deposition. Finally, the PANI/PPSU membrane sample was rinsed with DI water at a rate of 0.4 mL/s for 30 s to remove weakly bound PANI oligomers from substrate membrane. Formation of the PANI thin film on the PPSU membrane and the oxidative polymerization process are illustrated schematically in Figure 1.

### 2.4. Characterization

#### 2.4.1. Scanning Electron Microscopy (SEM)

SEM imaging was performed to examine the surface and cross-sectional morphology of the membranes using a SEM (JEOL, Tokyo, Japan). Membrane samples with areas of approximately 0.5 cm^2^ were taped to supports and sputtered with a Pt coating to a thickness of ~10 nm. The membranes were oriented perpendicularly to the incident electron beam to obtain cross-sectional images. The SEM images were collected at an accelerating voltage of 10 kV and a working distance of 6.4 mm at various magnifications.

#### 2.4.2. Atomic Force Microscopy (AFM)

The topography of the deposited PANI film was analyzed using a Nanoscope V MultiMode atomic force microscope (Veeco, San Jose, CA, USA). Scans were conducted at a frequency of 0.999 Hz using a Si tip in tapping mode. The surface roughness and structure were evaluated in a 5 μm × 5 μm area of the sample using the instrumental software (6.13).

#### 2.4.3. Contact Angle Measurements

The hydrophilicity of the membranes was quantified via the sessile drop method. Contact angles were measured using a goniometer (Atension, MAC 200, Biolin Scientific, Amsterdam, The Netherlands) equipped with a digital camera and image analysis software. Water droplets with volumes of 3 ± 1 µL were placed at five different places on the membrane surface using a microliter syringe. The profile of each water droplet relative to the membrane surface was recorded as a two-dimensional image using the digital camera. Each reported contact angle (*θ*) is the average of five measurements.

#### 2.4.4. Zeta Potential Measurements

The surface streaming potentials were measured using a SurPASS Electrokinetic analyzer (Anton Paar GmbH, Graz, Austria) to determine the zeta potentials of the membranes over a range of pH values. The zeta potential is an indicator of surface charge. The zeta potential of each membrane was measured after clamping the membrane in a cell with an adjustable gap. Titration was performed from pH 2.5 to pH 6 using a 0.01 mM KCl solution and 0.25 M HCl. The reported zeta potentials of the membranes were obtained from measurements at two locations on each sample.

### 2.5. Membrane Performance Assessment

#### 2.5.1. Experimental Setup and Water Filtration Procedure

A CF042 crossflow filtration system (Sterlitech, Kent, WA, USA) was used to evaluate membrane performance, water flux, and solute rejection. The active area of each membrane in the crossflow cell was 42 cm^2^. DI water was used to pressurize each membrane at 10 bar for 1 h, which allowed the water permeation flux to reach a steady state. After compaction, each membrane sample was mounted on the filtration cell assembly. The DI water flux was measured again under the same conditions for at least 1 h. For the final water flux measurements, the operative pressure was reduced to 6 bar. The flux was recorded as the mean of the 10 most recent water flux values. The permeated water was collected, and the DI water permeation flux was calculated using Equation (1).
*J*_w_ = *V*/(*A* × Δ*t*),(1)
where *J*_w_ is the water permeation flux through the membrane (L m^−2^ h^−1^), *V* is the collected permeate volume (L), *A* is the active membrane area (m^2^), and Δ*t* is the permeation time (h).

#### 2.5.2. Experimental Setup Used to Evaluate MB Dye Rejection

The DI water was drained from the storage feed tank, and the MB stock solution was added to assess rejection at a constant pressure of 6 bar. The permeate water was collected at intervals. Absorbance by MB was measured at 620 nm using a Cary 60 ultraviolet-visible (UV-VIS) spectrophotometer (Agilent Technologies, Santa Clara, CA, USA). Rejection (%) was then calculated using Equation (2).
*R* (%) = (1 − *C*_p_/*C*_f_) × 100,(2)
where *R* is the rejected amount of MB (%), *C*_p_ is the concentration of MB in the permeate, and *C*_f_ is the concentration of MB in the feed solution.

### 2.6. Antibacterial Activities and Anti-Adhesive Properties of the Membranes

#### 2.6.1. Antibacterial Activity

The antibacterial activities of the pristine PPSU and PANI-coated PPSU membranes against Gram-negative *E. coli* ATCC 25922 and Gram-positive *S. aureus* ATCC 25923 were evaluated using a standard plate counting technique. Luria Bertani (LB) broth was used to prepare each bacterial inoculum. The *E. coli* and *S. aureus* suspensions were added to separate wells in a 12-well culture plate. Pieces 2 cm^2^ in size were cut from each membrane, placed in the wells with the bacterial suspensions, and incubated at 37 °C. After six hours of incubation, 100 μL aliquots were collected from the wells and poured onto fresh LB agar plates. The plates were stored overnight in a bacteriological incubator at 37 °C. The same procedure was performed after 16 hours of incubation in the 12-well culture plate. The number of colonies on each plate was counted, and the percentage of growth inhibition (*B*_r_) was calculated using Equation (3).
*B*_r_ (%) = [(*A* − *B*)/*A*] × 100,(3)
where *A* is the number of colonies on the pristine membrane (control), and *B* is the number of colonies on the membrane coated with PANI.

#### 2.6.2. Effects of PANI on Bacterial Cell Morphology and Adhesion

SEM was used to analyze changes in the morphology and adhesion of bacterial cells grown on the PANI-coated and pristine PPSU membranes. The *E. coli* and *S. aureus* cultures were incubated with 2 cm^2^ pristine PPSU and PANI-coated PPSU membrane samples in 12-well culture plates at 37 °C for 12 h. After incubation, each membrane was removed from the plate and washed with phosphate buffered saline (PBS) to remove loosely attached cells. The membranes were then fixed using 4% glutaraldehyde and dehydrated in 30%, 50%, 70%, 90%, and 100% ethanol for 10 min each. The membranes were dried and coated with Au for SEM imaging at an accelerating voltage of 20 kV.

## 3. Results and Discussion

### 3.1. Membrane Morphology

Cross-sectional views of the unmodified PPSU and PANI-coated PPSU membranes are shown in Figure 2. The PANI thin film on the surface of the porous PPSU substrate is clearly visible in Figure 2b. The thickness of the PANI film was on the nanometer scale. The PANI film was dense, and its surface was free of defects. The PANI film had a uniform thickness, and it adhered tightly to the skeleton of the PPSU membrane.

Figure 3 shows the effect of the PANI film on the PPSU membrane structure (i.e., surface porosity-view and mean pore diameter). As shown in the results, the PANI/PPSU membrane film had a nanoporous structure with pores less than 5 nm in diameter (Figure 3b). Moreover, the top-surface texture parameters to quantify the magnitude of the peak material volume Vmp, core material Vmc, core void Vvc, and valley void Vvv were calculated by analyzed the SEM image with the ImageJ^®^ software and Mountains^®^ surface imaging software. It can be clearly seen in Figure 3a,b, Vmp (peak material volume) decreased significantly. On the whole, the deposition of the PANI film helped in diminishing irregularities, grooves, and macro voids over the surface of the PPSU membrane.

AFM was performed to examine the surface features of the membranes. The AFM results are shown in Figure 4. The two- and three-dimensional AFM images of the unmodified PPSU membrane in Figure 4a,b showed that its surface contained ridges and valleys. There were also large crater-like structures, deep grooves, small honeycomb-shaped micro-pits, and globular protrusions scattered across the membrane surface. However, the surface topography of the PANI-coated membrane was smoother and less wrinkled. The PANI-coated membrane had a smooth surface with an average roughness (R_a_) value of 3.15 nm (Figure 4d), which was notably smaller than that of the unmodified PPSU membrane (4.02 nm). Most of the ridges and valleys were filled in by the PANI film. The AFM results also showed that deposition of the PANI film assisted with the nanometer membrane surface preparation. The surface of the PANI film contained hemispherical nano pillars. These features were highly significant, because they facilitated water permeation and separation.

### 3.2. Membrane Hydrophilicity

The water contact angle is used to quantify the wetting behavior of membrane surfaces. Membranes with water contact angles below 90° are generally considered hydrophilic, meaning that water spreads easily across the surface. We found that pristine PPSU membranes were relatively hydrophobic with a water contact angle of 65° (Figure 5). The PANI-coated PPSU membrane also demonstrated hydrophilic behavior with a contact angle of 55°. The obtained results were in agreement with those reported previously in the literature. The decrease in the water contact angle due to deposition of the PANI film was consistent with the observations of other researchers. For instance, Huang et al. [34] reported that PANI membranes in their base form were quite hydrophobic. However, when the PANI was doped, the positive charges induced in the polymer chains and the negatively charged counterions increased hydrophilicity. Epstein et al. explained that the protonation of PANI doped with HCl gave rise to a polar conduction band, which led to a metallic state.

Doping increased the concentration of ionic charges in the conducting polymers, which indicated that intermolecular attraction between water and doped PANI would be stronger [35,36,37]. Doping has great potential for tuning the surface properties of PANI. It is therefore surprising that dopants, such as HCl and H_2_SO_4_, do not react chemically with the PANI polymer. Rather, they are in close proximity to the PANI polymer main chains. Depending on the type of dopant and its polarity, the presence of ionic charges on the PANI surface make it more hydrophilic. In addition, intermolecular hydrogen bonds formed between PANI and H_2_O, which strengthened intermolecular interactions between PANI and water. The combination of these factors made the surface of the PANI-coated PPSU membrane more hydrophilic [38,39].

### 3.3. Zeta Potentials of the Membrane Surfaces

The zeta potential is a measure of the electrical charge on a material surface, and its magnitude is an important measure of membrane fouling, wettability, and the rejection of charged and uncharged solutes. Zeta potential is plotted as a function of pH for both membranes in Figure 6. The zeta potentials were calculated from the streaming potentials, which were measured using an electrokinetic analyzer.

The unmodified PPSU membrane had a zeta potential (*ζ*) of −10.5 mV. The zeta potential of the PANI-coated PPSU membrane shifted from −10.5 to −1.7 mV as the pH increased from 2.5 to 7.5. This could be attributed to the presence of –NH groups in the positively charged PANI polymer chains on the modified membrane surface. Positively charged groups, such as amino groups, contribute to a decrease in the magnitude of *ζ*. In other words, they cause a positive shift in the zeta potential. The pH_iep_ values of the PANI-coated membrane and the unmodified PPSU membrane were 5.2 and 3.4, respectively. The PANI-coated PPSU membrane had a higher pH_iep_ value due to the large number of –NH groups in the cationic PANI polymer chains. 

### 3.4. Water Permeability and Rejection of the MB Dye

The contact angle on the unmodified PPSU membrane (65°) indicated that it was less hydrophilic than the PANI-coated PPSU membrane, which had a contact angle of 55°. The more hydrophilic nature of the PANI-coated membrane led to stronger interactions with water, which favored an increase in the permeability of the PANI-coated PPSU membrane to water (Figure 7).

The increased permeability could be attributed to the nanometer-scale thickness of the PANI film, which significantly decreased the resistance of the membrane to flowing water. These results were consistent with those reported previously in the literature. Ahmad et al. reported on PANI nanoparticles that enhanced water permeability and improved the anti-fouling performance of membranes, which was due to an increase in hydrophilicity. Moreover, in the literature, researchers demonstrated that PANI–H_2_O complexes formed as a result of H–O----H–N and C–N----H–O intermolecular hydrogen bond formation [38,39,40]. This strengthened the interaction between PANI and water molecules, which resulted in high water permeation.

In terms of membrane rejection efficacy, the performance of the PANI-coated PPSU membrane was excellent. It rejected over 90% of the MB dye (Figure 8). These results could be interpreted in two ways. The presence of PANI molecules on the membrane surface led to a positive shift in the zeta potential from −10.5 to −1.7 mV, which increased the repulsive force between PANI molecules and the MB dye. The size exclusion mechanism was more dominant based on the porosity of the membrane surface [41,42,43]. The sizes of the pores in the PANI-coated PPSU membrane were in the nanoscale range, which increased rejection efficacy. Overall properties of developed PANI-coated PPSU membrane shown in Table 1.

### 3.5. Antibacterial Activities of the Membranes

The antimicrobial activity of pristine PANI and PANI nanocomposites, including PANI-Ag, PANI-Au, PANI-Ag-Au [44,45], PANI/PVA/Ag [46], PANI/Ag-Pt, PANI/Pt, PANI/Pt-Pd [47,48], CoFe_2_O_4_/PANI/Ag [49], Cu/PANI [50], and PBT-g-MA/PANI [51], against streptococci, *Bacillus*
*subtilis*, staphylococci, *E. coli*, *Klebsiella* sp., *Pseudomonas*
*aeruginosa*, *Salmonella*, and *Shigella*, is well documented in the biomedical literature. However, few studies have focused on biofouling of PANI composite membranes and their application for ultrafiltration, desalinization, and wastewater treatment. Razali et al. [52] reported that PANI nanoparticles blended with PES membranes and improved both biofouling performance and antibacterial activity. To the best of our knowledge, no reports on the antibacterial activities of PANI-coated PPSU membranes against *E. coli* and *S. aureus* have been published. *E. coli* and *S. aureus* were used in this study, because these strains are generally used as biological indicators to assess water contamination [40,50]. 

PANI nanoparticles served as hydrophilic polymeric materials in this study, and we deposited them on a PPSU membrane. A new membrane formed as PANI was deposited on the PPSU support, which had enhanced biofouling resistance and antimicrobial properties for water purification. The antibacterial activities of a pristine PPSU membrane sample and a PANI-coated PPSU membrane sample against *E.*
*coli* are shown in Figure 9. The activities of both types of membrane against *S. aureus* are shown in Figure 10. There were many more bacterial colonies on the pristine PPSU membranes, which did not exhibit antibacterial activities.

The PANI-coated PPSU membrane had superior antibacterial activity, which inhibited attachment and growth of the bacteria. Similar results were reported for PES/PANI membranes against *E. coli* and *Bacillus* sp. [52]. Based on our calculated growth inhibition (*B*_r_) values, the PANI-coated PPSU membrane reduced the growth of *E. coli* by 63.5% within six hours. Within 16 hours, the PANI-coated PPSU membrane reduced *E. coli* growth by 95.2%. The *B*_r_ values of the PANI-coated PPSU membrane against *S. aureus* after six and sixteen hours of exposure were 70.6% and 88.0%, respectively.

### 3.6. Effects of PANI Deposition on the Morphology and Adhesion of Model Bacterial Cells

SEM analysis was performed to investigate the effects of the PANI film on the morphology and adhesion of *E. coli* and *S. aureus* cells after 12 h and 24 h of incubation. The SEM images of pristine PPSU membranes incubated with *E. coli* and *S. aureus* showed smooth surfaces with compact and intact cell walls and membranes (Figure 11). In contrast, the surface morphologies of both *E. coli* and *S. aureus* cells grown on PANI-coated PPSU membranes were irregular and abnormal. PANI restricted the attachment and colonization of the bacteria to a greater extent (Figure 11). The cell walls of *E. coli* were severely damaged with a large number of visible high degree indentations and cavities. Collapse of the indentations and cavities ruptured the cytoplasmic membranes, resulting in leakage of the cytoplasmic contents and solutes. Bogdanović et al. [50] reported that the death of *E. coli* cells following treatment with Cu/PANI was caused by the direct rupture of bacterial cell membranes.

It has been reported that the effect of PANI coatings on Gram-positive *S. aureus* differs from their effect on *E. coli* cells. The *S. aureus* cells grown on pristine PPSU were typically spherical in shape with intact cell walls and no damage. However, cells on the PANI-coated PPSU membrane were damaged. Their surfaces were rougher and contained cavities. Although the cell membranes did not collapse, significantly fewer cells grew on the PANI-coated PPSU membrane. The colony counting results and SEM data clearly showed that the PANI-coated PPSU membrane was more effective against Gram-negative *E. coli* than Gram-positive *S. aureus*. Similar results were reported by Dhivya et al. [53]. This may be have been due to differences between the cell wall structures of the bacteria. The cell walls of Gram-positive bacteria are comprised of complex, rigid, and thick (20–80 nm) peptidoglycan layers that provide protection and few anchoring sites for PANI chains [53]. In contrast, the cell walls of Gram-negative bacteria consist of thin (7–8 nm) peptidoglycan layers and negatively charged lipopolysaccharides that attract PANI. Dhivya et al. [53] reported that PANI chains with positively charged amino groups bonded and anchored to negatively charged lipopolysaccharides and damaged the cell membranes, which was due to changes in the potential gradient across the ion channels [53]. 

The exact mode of PANI activity is not clear, but it has been reported that antimicrobial activity against various types of bacteria depends on surface hydrophilicity [54], electrostatic interactions [55], the presence of amino groups [51,56], direct physical contact, and high molecular weight. PANI increases the interfacial area between bacteria and nanocomposites and the length of the polymer chains. PANI and PANI nanocomposites attach and anchor to bacterial cell walls, which disrupts their integrity and permeability. Gizdavic-Nikolaidis et al. [57] proposed that the disruption of cell walls and leakage of critical cytoplasmic contents caused Gram-positive and Gram-negative bacteria to die. This was attributed to direct cationic binding of PANI to components in the cell walls due to electrostatic interactions. 

## 4. Conclusions

In this study, a nanofiltration membrane was fabricated by depositing a PANI film on a PPSU substrate membrane. The PANI coating was a potent separation enhancer with powerful antimicrobial activity. The deposited PANI film led to significant changes, including a fine-tuned nanometer pore size and rejection of over 90% of MB dye. The PANI coating also enhanced surface properties, which was indicated by greater hydrophilicity and a shift in the zeta potential from −10 to −1.7 mV. The PANI-coated membrane displayed a permeance of 20 L·m^−2^·h^−1^·bar^−1^. This was because the ~200 nm thick PANI film provided low-resistance transport channels, which enhanced membrane permeability. The PANI film also made the PPSU membrane more resistant to bacteria and significantly inhibited the growth of *E. coli*.

## Figures and Tables

**Figure 1 membranes-11-00025-f001:**
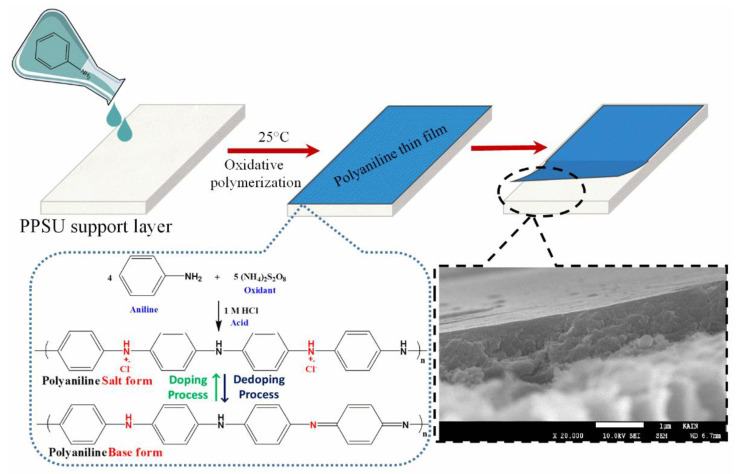
Schematic diagram of PANI thin film preparation via oxidative polymerization.

**Figure 2 membranes-11-00025-f002:**
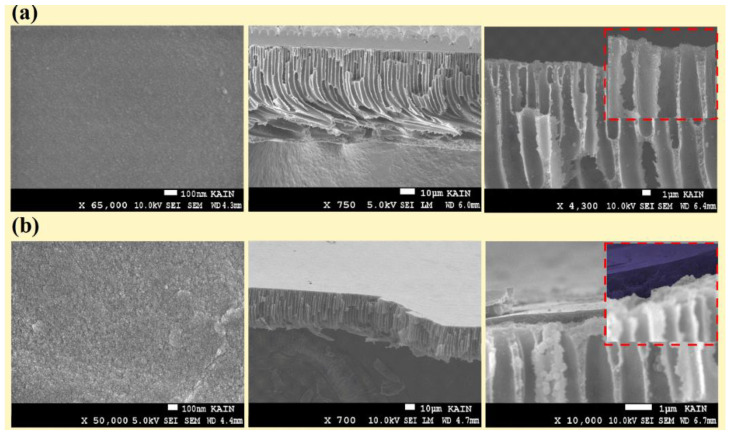
Cross-sectional scanning electron microscope (SEM) images of (**a**) the Poly (phenyl sulfone) (PPSU) support and (**b**) the PANI-coated PPSU membrane.

**Figure 3 membranes-11-00025-f003:**
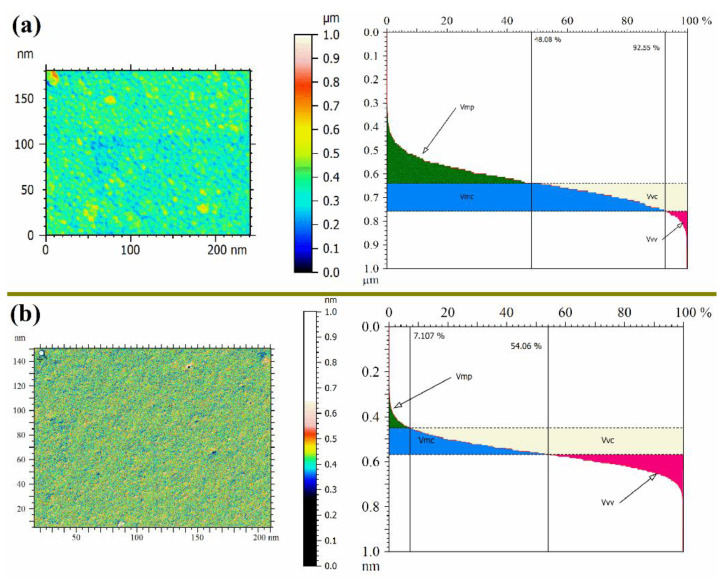
The surface porosity-view of the membranes (**a**) PPSU support, (**b**) PANI-deposited PPSU and denoted as peak material volume (Vmp), valley void volume (Vvv), core material volume (Vmc), and core void volume (Vvc).

**Figure 4 membranes-11-00025-f004:**
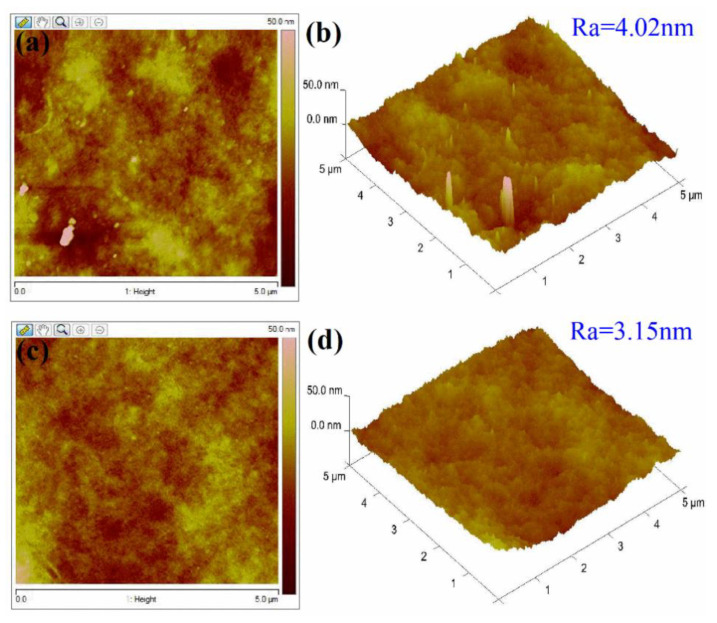
Representative 2D and 3D AFM images of the surfaces of (**a**,**b**) the PPSU support and (**c**,**d**) the PANI-coated PPSU membrane.

**Figure 5 membranes-11-00025-f005:**
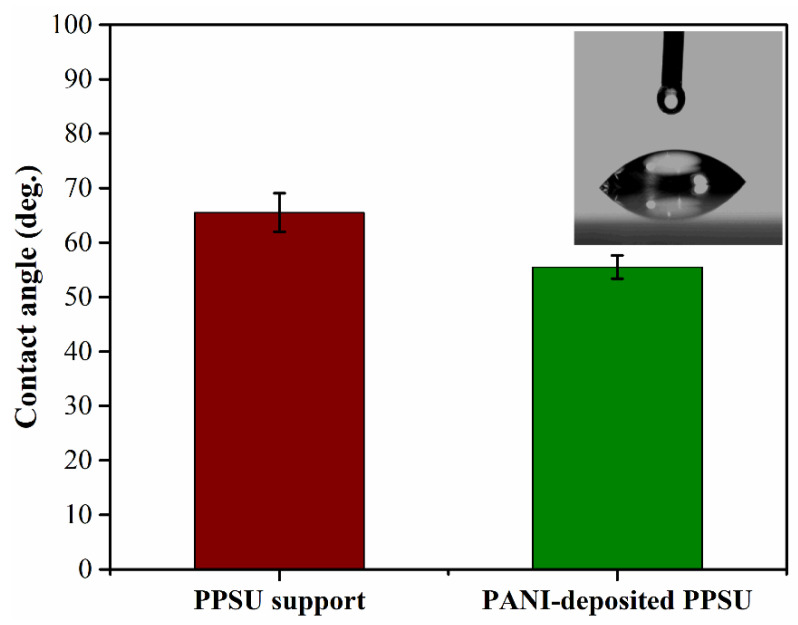
Water contact angles of the membranes.

**Figure 6 membranes-11-00025-f006:**
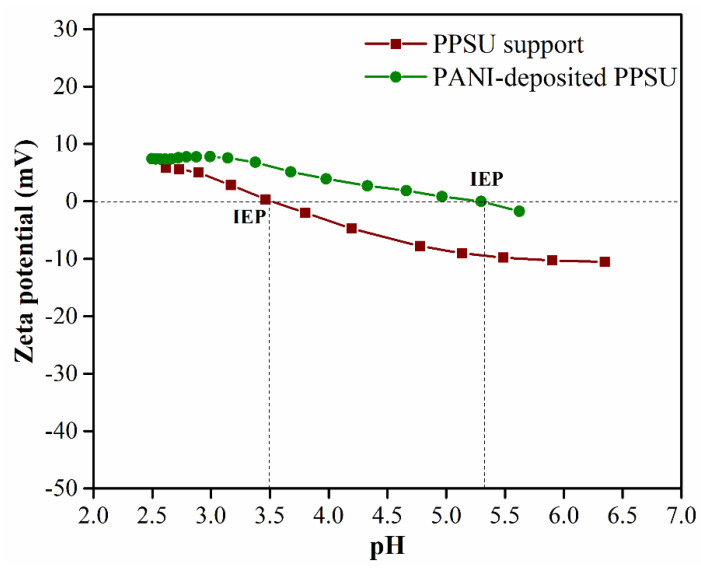
Zeta potentials of the PPSU support membrane and the PANI-coated PPSU membrane as a function of pH.

**Figure 7 membranes-11-00025-f007:**
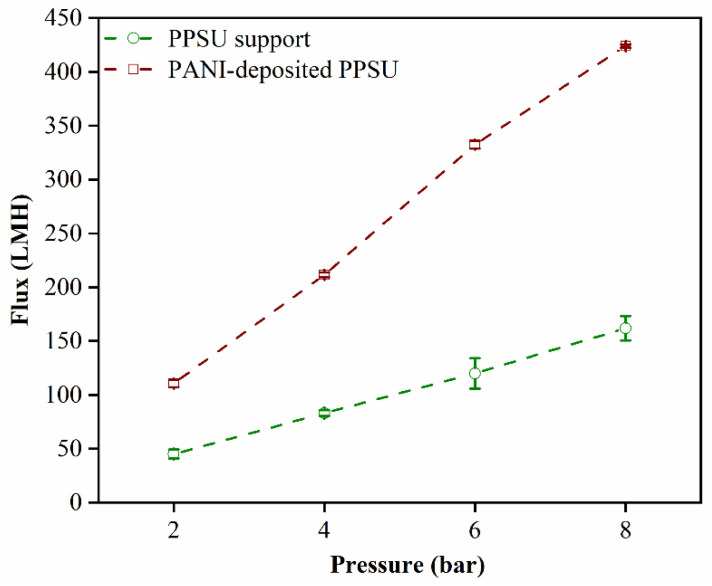
Permeability of the membranes to pure water as a function of pressure.

**Figure 8 membranes-11-00025-f008:**
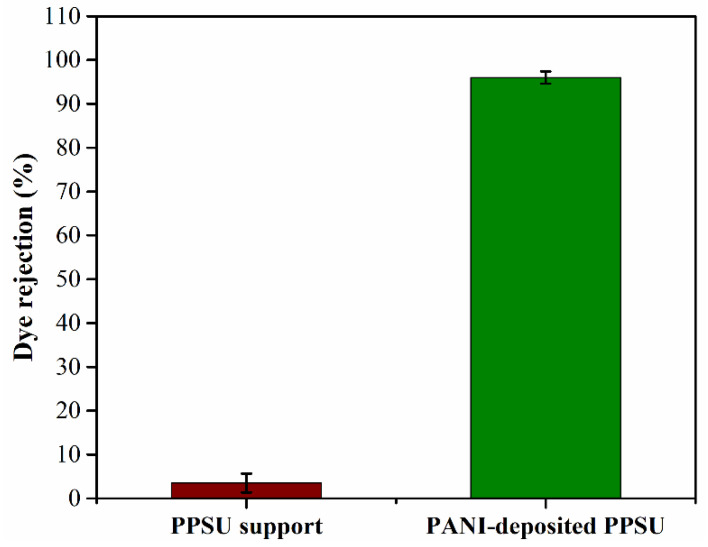
Rejection of the MB dye by the PPSU support membrane and the PANI-coated PPSU membrane.

**Figure 9 membranes-11-00025-f009:**
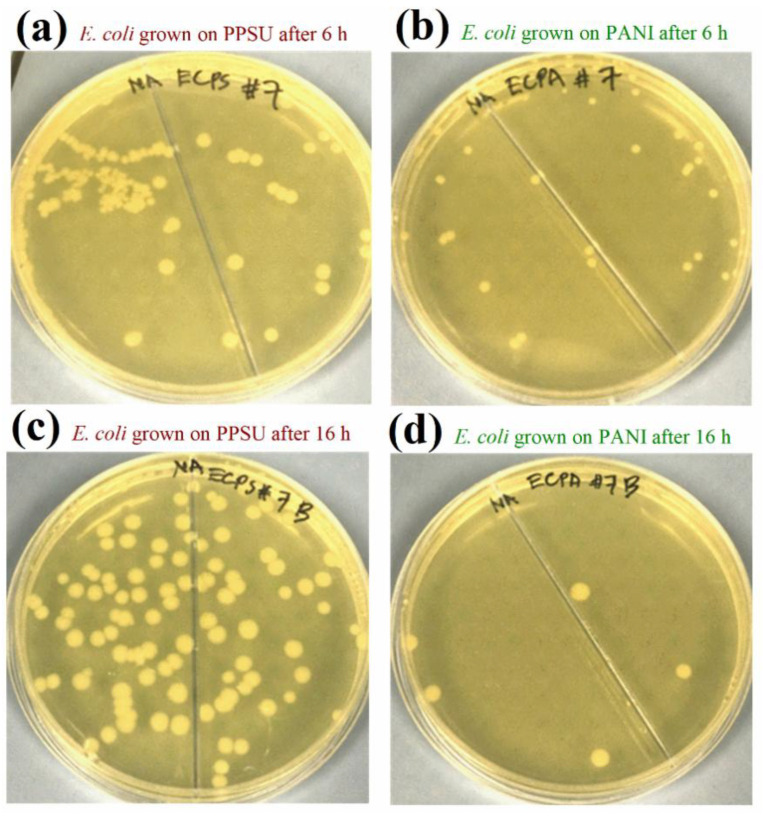
Antimicrobial properties of pristine PPSU and PANI-coated PPSU membranes assessed by plate counting. Plates incubated after the growth of *E. coli* on a pristine PPSU membrane for (**a**) 6 h and (**c**) 16 h. Plates incubated after the growth of *E. coli* on a PANI-coated membrane for (**b**) 6 h and (**d**) 16 h.

**Figure 10 membranes-11-00025-f010:**
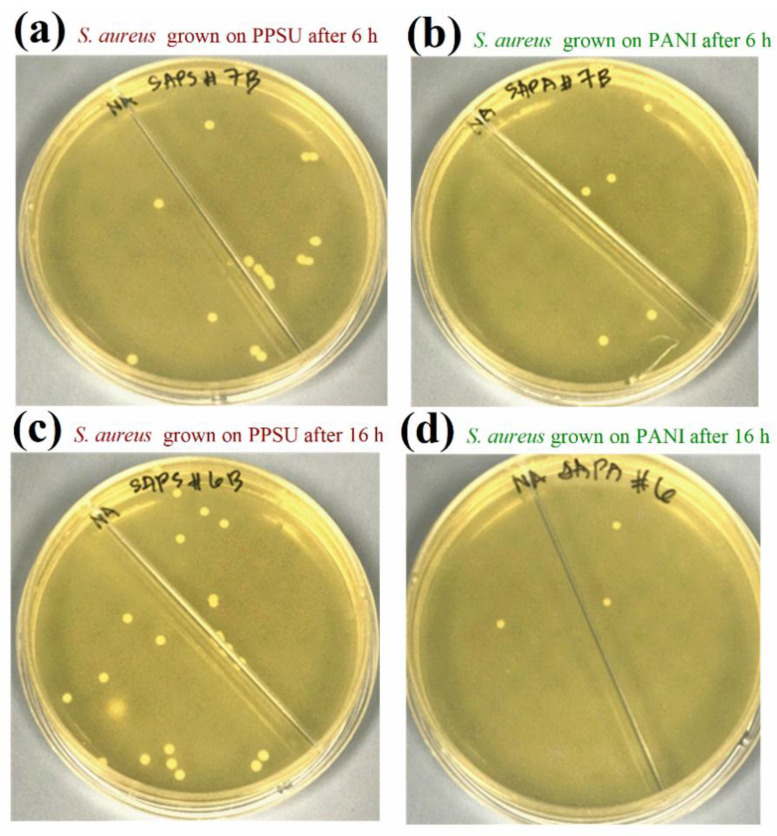
Antimicrobial properties of pristine and PANI-coated PPSU membranes assessed by plate counting. Plates incubated after the growth of *S. aureus* on a pristine PPSU membrane for (**a**) 6 h and (**c**) 16 h. Plates incubated after the growth of *S. aureus* on a PANI-coated PPSU membrane for (**b**) 6 h and (**d**) 16 h.

**Figure 11 membranes-11-00025-f011:**
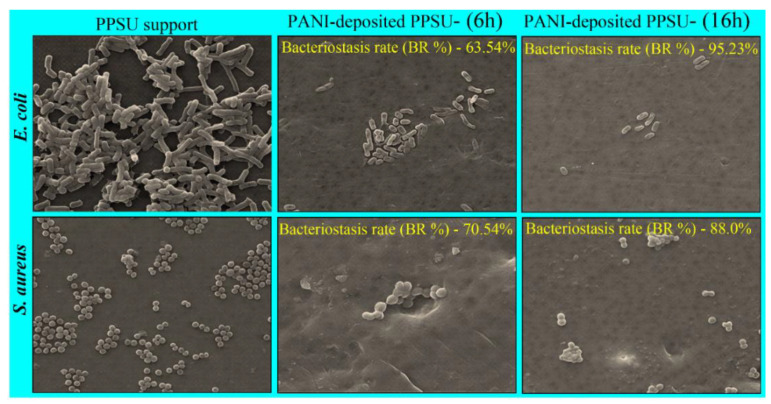
SEM images of features in at 5 kV, 1000 X, 6.5 WD collected to evaluate adhesion and the morphologies of *E.*
*coli* and *S. aureus* cells on pristine PPSU and PANI-coated PPSU membranes after incubation for 6 h and 12 h.

**Table 1 membranes-11-00025-t001:** Properties of the prepared membranes in this study.

Membranes	Isoelectric Point (pH_iep_)	Water Permeability (L·m^−2^·h^−1^·bar^−1^)	Dye Rejection (%)
PPSU membrane	3.4	20.3 ± 3	3 ± 0.5
PANI-coated PPSU membrane	5.2	53.5 ± 5	96 ± 2

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
