# Peer review of "Dye Separation and Antibacterial Activities of Polyaniline Thin Film-Coated Poly(phenyl sulfone) Membranes"

_membranes, 2020, doi:10.3390/membranes11010025_

Round 1
Reviewer 1 Report
The revised manuscript could be accepted by membrane
Author Response
The authors highly appreciate the valuable comments of the referee on the manuscript.
Reviewer 2 Report
The manuscript by Alam and co-workers describes the preparation of PANI coated membranes. The work has some merits and the fits the scope of the journal. However, there are several minor and major issues to be addressed before further consideration.
1, What do the authors mean by ‘PANI has been investigated primarily as a bulk powder’? PANI is very well know, and investigated polymer.
2, The authors claim that they prepared a thin film but the thickness of the film is not measured. The precise determination of the thickness of the coated layer is necessary, as reported in similar works.
3, What percentage of the used monomer ended up being incorporated into the thin film upon polymerization? This is important to report as this is an important cost and feasibility aspect.
4, The objective of the work is to achieve hydrophilicity, which the authors emphasized but the water contact angle with PANI is almost the same as without it (Figure 5). The authors should explain the reasons, and what has been achieved.
5, The Section 2.1 on materials should include the purity, grade, molecular weight etc of the materials used.
6, The authors should mention the extensive study on hydrophilic coatings on membrane supports for tuning separation performance in the introduction (10.1021/acsapm.8b00161; 10.1016/j.seppur.2019.116412).
7, Some critical aspects on the work in general should be included, where possible pros and cons of the applied materials and approaches should be mentioned. What are the limitations and drawbacks of the proposed methodologies?
8, Figures 3, 11 SEM images need to have legible scale bars. Currently it is impossible to interpret the images as the scale cannot be determined.
9, At the end of section 3.4, the authors should add a table comparing the flux/permeance and dye rejections with broad examples found in the literature (10.1016/j.memsci.2018.09.001; 10.1039/D0TA08194A; 10.1016/j.memsci.2020.118241).
Author Response
Point-by-Point Response to Reviewer
Query (1): What do the authors mean by ‘PANI has been investigated primarily as a bulk powder’? PANI is a very well know and investigated polymer.
Response: Thank you very much for the valuable comment. It is well-documented in the literature that polyaniline, i.e., PANI can be prepared as power form, and also can be prepared film form on a variety of substrates such as glass, metals, etc. For membrane materials science and technology, PANI has been used in the literature mostly as power form, i.e. nanoparticles to develop composite, nanocomposites, and blend membranes. PANI, however, has not been much investigated as film form, although it has outstanding properties such as switchable wettability due to doping-de doping phenomena, thermal, chemical stability, which all are beneficial to develop advanced membranes with antifouling, antimicrobial fouling properties. That is why the authors stated that PANI has been investigated primarily as a bulk powder.
Query (2): The authors claim that they prepared a thin film but the thickness of the film is not measured. The precise determination of the thickness of the coated layer is necessary, as reported in similar works.
Response: It is a nice comment. The authors agree with the reviewer's comment on the thickness of the deposited PANI thin film. For thickness measurement, in the study, no extra tool was used because approximately thickness was determined by SEM result. As it can be seen clearly in the SEM image highlighted in blue color, “bar scale” is 1 micro-meter, i.e. 1000 nm. Form this scale bar, the thickness of the deposited thin film was calculated, which is almost ~250 nm.
Query (3): What percentage of the used monomer ended up being incorporated into the thin film upon polymerization? This is important to report as this is an important cost and feasibility aspect.
Response: Thank you for the comment, and this is important to report as this is an important cost and feasibility aspect. The used aniline monomer percentage was 4 wt. %.
Query (4): The objective of the work is to achieve hydrophilicity, which the authors emphasized but the water contact angle with PANI is almost the same as without it (Figure 5). The authors should explain the reasons, and what has been achieved.
Response: It is an excellent comment, and thank you for this notice. The authors agree with the reviewer's notice, “the objective of the work is to achieve hydrophilicity, which the authors emphasized but the water contact angle with PANI is almost the same as without it (Figure 5).” No doubt, with the deposition of PANI film, changes in hydrophilicity were not much significant. But there are many other significant changes observed such as:
The zeta potential of the PANI-coated PPSU membrane shifted from −10.5 to −1.7 mV as the pH increased from 2.5 to 7.5, that is we can say the membrane is shifting cationic nature, which is beneficial for repelling cationic dyes. As it was observed in the results, shown in Fig. 8. PANI deposited membrane rejected over 90% of the MB dye. This is because, with the deposition of PANI, the membrane surface led to a positive shift in the zeta potential from −10.5 to −1.7 mV, which increased the repulsive force between PANI molecules and the MB dye.
Secondly, as it can be seen that, with the deposition of PANI, the size exclusion mechanism was more dominant based on the porosity of the membrane surface. The sizes of the pores in the PANI-coated PPSU membrane were in the nanoscale range, which increased rejection efficacy.
Query (5): The Section 2.1 on materials should include the purity, grade, molecular weight etc of the materials used.
Response: In the revised manuscript, we added materials purity, grade, molecular weight.
Query (6): The authors should mention the extensive study on hydrophilic coatings on membrane supports for tuning separation performance in the introduction (10.1021/acsapm.8b00161; 10.1016/j.seppur.2019.116412).
Response: In the revised manuscript, suggested literature is added to enrich the content of the introduction part.
Query (7): Some critical aspects of the work, in general, should be included, where possible pros and cons of the applied materials and approaches should be mentioned. What are the limitations and drawbacks of the proposed methodologies?
Response: In perusing our investigation, we observed that PANI is a promising material for the development of membranes with antifouling properties, on the other hand, there is some limitation of PANI membranes, such as surface wettability is changed with varying the pH value. During membrane fold, PANI film may crack if it is that of high thickness more than 500 nm.
Query (8): Figures 3, 11 SEM images need to have legible scale bars. Currently, it is impossible to interpret the images as the scale cannot be determined.
Response: Thank you for the suggestion, in the revised manuscript, is done.
Query (9): At the end of section 3.4, the authors should add a table comparing the flux/permeance and dye rejections with broad examples found in the literature (10.1016/j.memsci.2018.09.001; 10.1039/D0TA08194A; 10.1016/j.memsci.2020.118241).
Response: In the revised manuscript, it is done.

Reviewer 3 Report
1.Describe, please, more clearly what is the novelty of the study?
2. What was molecular weight of PVP used?
3. What was the concentration of PPSU in the casating solution?
4.The contact angle of the membrane selective layer was determined via the sessile drop method. Was there a water drop soaking inside the membrane pores with time? Maybe, a bubble attached technique is better to apply.
5.Authors state that " The more hydrophilic nature of the PANI-coated membrane led to stronger interactions with water, which slightly reduced the permeability of the PANI-coated PPSU membrane to water" (Lines 280-282). This is not correct as more hydrophilic membranes are known to be more permeable to water.
6. Authors state that " This could be attributed to the nanometer-scale thickness of the PANI film, which significantly decreased the resistance of the membrane to flowing water" (Lines 289-290). This is not correct. If the permeability decreased the resistance of membrane increased.
7. Authors state that "A marginal decrease in water flux resulted from the combined effects of lower surface roughness and the greater porosity of the PANI film, which was reflected in our AFM images". This is not correct as low surface roughness can not decrease water permeability. It is not clear how can the deposition of PANI film on the surface of the serlective layer increase membrane porosity? It is more likely that water permeability decrease is attributed to the partial pore blockage or narrowing due to formation of additional layer on the surface.
8. Figure 11. What was the time of incubation for prostine PPSU membrane?
Author Response
Point-by-Point Response to Reviewer
Query (1): Describe, please, more clearly what is the novelty of the study?
Response: Usually, polyphenylsulfone (PPSU) membrane is widely used in ultra-filtration applications. In the current study, a thin PANI film was deposited on the surface of a PPSU substrate membrane to extend PPSU membrane separation application, particularly in the field of nanofiltration. To the best author's knowledge, PANI has been investigated widely as a powder to develop composite, nanocomposite, or blend PPSU membranes. However, PANI has not been utilized to cast films on PPSU substrate membranes to enhance their separation properties, although it has great potential to develop nanopores on the membrane surfaces. PANI as a film form has outstanding properties such as switchable wettability due to doping-de doping phenomena, thermal, chemical stability, which all are beneficial to develop advanced membranes with anti-fouling, antimicrobial fouling properties.
Query (2): What was the molecular weight of PVP used?
Response: In the revised manuscript, the molecular weight of PVP is mentioned.
Query (3): What was the concentration of PPSU in the casting solution?
Response: In the revised manuscript, the concentration of PPSU in the casting solution is already given in the membrane fabrication section.
Query (4): The contact angle of the membrane selective layer was determined via the sessile drop method. Was there a water drop soaking inside the membrane pores with time? Maybe, a bubble attached technique is better to apply.
Response: Thank you for a good suggestion. A bubble attached technique can apply. Yes, in the sessile drop method, there was a water drop soaking inside the membrane pores with time. Mostly, the sessile drop method is used to make direct measurements of the contact angle to determine preferential wetting of a given membrane by water. With the experiment, Water droplets were placed on the membrane surface using a microliter syringe then, left for 20 sec to let the membrane soak water. After that, the profile of each water droplet relative to the membrane surface was recorded using the digital camera.
Query (5): Authors state that " The more hydrophilic nature of the PANI-coated membrane led to stronger interactions with water, which slightly reduced the permeability of the PANI-coated PPSU membrane to water" (Lines 280-282). This is not correct as more hydrophilic membranes are known to be more permeable to water.
Response: First of all, thank you very much for the comments. In the revised manuscript, the discussion of water permeability is modified. Now it is corrected.
Query (6): Authors state that " This could be attributed to the nanometer-scale thickness of the PANI film, which significantly decreased the resistance of the membrane to flowing water" (Lines 289-290). This is not correct. If the permeability decreased the resistance of membrane increased.
Response: In the revised manuscript, the discussion of water permeability is corrected and added in revised manuscript.
Query (7): Authors state that "A marginal decrease in water flux resulted from the combined effects of lower surface roughness and the greater porosity of the PANI film, which was reflected in our AFM images". This is not correct as low surface roughness can not decrease water permeability. It is not clear how can the deposition of PANI film on the surface of the selective layer increase membrane porosity? It is more likely that water permeability decrease is attributed to the partial pore blockage or narrowing due to the formation of an additional layer on the surface.
Response: Thank you very much for a nice suggestion, it really improve the quality of our manuscript. Hence, the suggested things are included in the revised manuscript.
Query (8): Figure 11. What was the time of incubation for the pristine PPSU membrane?
Response: Pristine PPSU and PANI-coated PPSU membranes incubation time is 6 h and 12 h.
Round 2
Reviewer 2 Report
The authors have done a thorough revision but some minor points remained to correct before publication:
1, Sale bars in Figure 11 should be added for all panels as the SEM images cannot be interpreted without them.
2, All the captions for figures that contain error bars should briefly mention how these errors were derived.
3, Both the quotient (“x/y”) and negative exponent (“x y-1”) formats are used in the manuscript for units. Either of them should be used consistently, preferably the negative exponent format, which is recommended by the IUPAC.
4, Reference #32 has no page numbers but an article number (66). References #40 (pages 3585–3601), #42 (article number 118241), #43 (pages 24445-24454), #54 (pages 4481–4494) are incomplete.
Reviewer 3 Report
Comments are adressed in a proper way. I suggest go accept the manuscript for publication.